# A Very Rare Case of Colosalpingeal Fistula Secondary to Diverticulitis: An Overview of Development, Clinical Features and Management

**DOI:** 10.3390/medicina56090477

**Published:** 2020-09-17

**Authors:** Natalia Darii Plopa, Nicolae Gica, Marie Gerard, Marie-Cécile Nollevaux, Milenko Pavlovic, Emil Anton

**Affiliations:** 1Department of Gynecology, CHU de Charleroi, 6000 Charleroi, Belgium; plopa_nati@yahoo.com; 2Carol Davila University of Medicine and Pharmacy, 020021 Bucharest, Romania; 3Department of Radiology, CHU Dinant Godinne|UCL Namur, 5530 Yvoir, Belgium; marie.gerard@uclouvain.be; 4Department of Pathology, CHU Dinant Godinne|UCL Namur, 5530 Yvoir, Belgium; marie-cecile.nollevaux@uclouvain.be; 5Department of Gynecology, Faculty of Medicine, Pontificia Universidad Católica de Chile, 833-0073 Santiago, Chile; mpavlovicv@gmail.com; 6University of Medicine and Pharmacology Gr T Popa, 700115 Iasi, Romania; emil.anton@yahoo.com

**Keywords:** colosalpingeal fistula, enterotubal fistula, diverticular fistulation, diagnosis, hysteroscopy management

## Abstract

Background: Colosalpingeal fistula is a rare complication secondary to diverticular disease. The pathogenesis is still not clearly understood. We present the case of a colosalpingeal fistula and a review of the management of this pathology. Case report: A 69-year-old patient with uncomplicated diverticular disease was referred to our department for recurrent vaginal discharge. The clinical examination was unremarkable, hysteroscopy revealed the presence of air in the uterine cavity in the absence of a uterine fistula. A preliminary diagnosis of colosalpingeal fistula was made and was confirmed by computed tomography (CT) scan and hysterosalpingography. A one-stage surgery via laparotomy was successfully performed with remission of the symptoms. Conclusion: Colotubal fistula is a rare complication resulting from intestinal diverticular disease. The purpose of this paper was to emphasize the presence of a rare, but serious complication occurring in diverticular disease with atypical symptoms and one-stage surgery treatment.

## 1. Introduction

The colotubal (or colosalpingeal or salpingo-intestinal) fistula occurs in 2% of the cases with fistulas secondary to diverticular disease [1]. The initial symptoms of this disease are not specific, but this complication must be suspected in patients with persistent or recurrent vaginal discharge and a history of diverticular disease. The specific diagnostic tests for this condition are computed tomography (CT) scan and hysterosalpingography. The management options depend on the patient´s age and desire to preserve fertility. We report a case of a menopausal women with known diverticular disease and colosalpingeal fistula, managed in our gynecology department, with a review of the literature.

## 2. Case Report

A 69-year-old postmenopausal caucasic woman with previous two pregnancies and two deliveries was referred to our department for a vaginal recurrent leucorrhoea which lasted for over one year despite multiple vaginal local treatments. Her past medical and surgical histories revealed two episodes of diverticulitis managed with antibiotics, appendectomy, and breast cancer treated with breast-conserving surgery followed by radiotherapy and hormonotherapy with complete remission of the disease. Unlike other cases from the literature reports, our patient presented with gynecological problems and her only symptom was the vaginal discharge, the patient did not complain about abdominal pain, fever, or other symptoms. The clinical examination was unremarkable, and vaginal swab revealed the presence of *Escherichia coli* and *Streptococcus constellatus*. The transvaginal ultrasound examination showed a heterogeneous intrauterine collection. Hysteroscopy was performed as an additional exam. The presence of air in the uterine cavity in the absence of uterine fistula or neoplasic process made us suspect the tubo-intestinal fistula. The endometrial biopsy was compatible with endometritis secondary to *E. coli*. To confirm the diagnosis, CT scan and hysterosalpingography were performed and the communication between the tube and sigmoidal diverticulum was visualized (Figure 1 and Figure 2).

Based on the medical history, age, and symptoms, elective surgical treatment was performed by gastrointestinal surgeons. This was a one-stage surgery with sigmoidal and concomitant left salpingo-oophorectomy and primary anastomosis. Intraoperative, the affected area of the colon was densely adherent to the left fallopian tube. Inflammation was present in the proximal rectum, therefore dissection was extended more distally and a low anterior resection with primary anastomosis was performed. The colon and fallopian tubes were successfully separated using finger fracture technique. Pathological examination confirmed the diverticular disease associated with neutrophilic cryptitis and serosal inflammation. The 6.5 cm length fallopian tube was characterized by a dilated lumen. At the microscopic level, the fallopian tube showed evidence of chronic and subacute salpingitis whereas the peritubal environment revealed some stercoral debris mixed with extensive abscessation beaches (see Figure 3A,B). There were no intraoperative complications. The patient had a favorable evolution during hospitalization and was discharged home on day 7 post surgery. One month later, she remained asymptomatic.

## 3. Discussion

Diverticular disease is becoming more common, affecting up to 71.4% of the population over 80 years [2]. Literature reports an increase from 49,000 cases of diverticular disease in 2000 to more than 70,000 in 2006, the complicated diseases almost doubled between 1990 and 2005 [3]. Many conditions of diverticular disease are described, from asymptomatic form to complicated aspects such as inflammation and fistulation to the surrounding burden organs. The incidence of enteric fistula in diverticular disease is about 2–4%, reaching up to 20% in patients with history of surgical treatment of diverticular disease [4]. Spontaneous colosalpingeal fistula secondary to diverticulosis is very rare (only 10 cases reported in the literature) and may appear even during pregnancy [5,6]. The first case of sigmoidotubal fistula was described in 1956 [7]. The prevalence of colotubal fistula is about 2% [1].

The colosalpingeal fistula is rare due to the tubal obstruction secondary to inflammatory process at the tubal level. Although the pathogenesis is not fully understood, some authors hypothesized that chronic inflammation is involved in the development of this pathology. The adhesions between intestinal wall and the fallopian tube may occur during an acute episode of diverticulitis, resulting in necrosis and fistula formation. This pathogenesis may explain the colosalpingeal fistula secondary to diverticulitis or Crohn’s disease [8]. Fistula can occur also due to a diverticulum perforation with local abscess formation and weakening of adjacent structures [9]. Moreover, the neoplastic fistula is formed following the same mechanism of inflammation, epithelial necrosis, and destruction associated with ulceration [10].

Colosalpingeal fistula occurs in the left tube in 93% of the cases, with the right tube affected in only 7% of the cases, due to cecal diverticulosis [11]. Until now, only one case of bilateral salpingocolic fistula was reported in the literature [12].

Colotubal fistulas are uncommon and difficult to diagnose. Clinical manifestations may be nonspecific, and the symptoms appear later in the evolution of the disease. Although clinical manifestations may vary, the most frequent symptom is persistent or recurrent vaginal discharge with the presence of enterocolic germs at the vaginal swab. Gynecologic examination is mandatory to exclude vaginal fistula and verify vaginal integrity. Endovaginal ultrasonography examination may indicate the presence of an intrauterine collection and may rarely reveal a parauterine collection or fistulous trajectory. Uterine cavity exploration by hysteroscopy should be performed to exclude the presence of a uterine fistula or a fistulized endometrial neoplasia. The presence of air in the uterine cavity associated with the absence of uterine communication or cancerous process may suggest the presence of a colosalpingeal fistula. The endometrial biopsy may reveal the presence of granulation tissue and exclude a neoplasic process.

CT scan is the gold standard test for the diagnosis of abdominal and pelvic organs fistulas [13]. The sensitivity and specificity of the CT scan for the diagnosis of colorectal fistula is 88% and 100%, respectively [11]. The presence of adnexal gas, adnexal fluid, intraabdominal gas, or intraabdominal collections associated with enlarged inflammatory adnexa was described in the cases of the colosalpingeal fistula [14]. The sensitivity and the specificity of CT scan and hysteroscopy for the diagnosis of colosalpingeal fistula remains to be established due to the low incidence of this pathology.

The hysterosalpingography is the imaging method with the highest sensitivity for the diagnosis of colosalpingeal fistula [15]. Colonoscopy helps to identify bowel pathology that caused an enteral fistula in 8.5–55% of cases [16].

Colosalpingeal fistula can also occur as a consequence of other conditions, different from diverticular disease, such as tuberculosis [17,18], appendicitis, or pelvic inflammatory disease [19,20]; primary tubo-ovarian abscess; endometriosis; and neoplasic conditions.

The treatment is a surgical one, only one case has been shown to resolve spontaneously [21], and it depends on the age of the patient, the medical and surgical history, and especially the desire of fertility preservation. The recommended surgical treatment is the resection of the affected bowel segment with primary anastomosis and resection of the complete adjacent organ or only the affected part of it. The primary intestinal anastomosis is possible in the absence of sepsis or severe malnutrition [22]. Meticulous treatment of the fistula is required if a conservative surgical approach is needed. Radical surgical treatment, such as Hartmann´s opereation, should be reserved only for emergencies, and secondary anastomosis should be performed once the acute inflammation has resolved. In the presence of acute diverticulitis, a preoperative medical treatment is preferred to decrease the risk of postoperative complications [23].

The recommended surgical treatment for colosalpingeal fistula in postmenopausal women is total hysterectomy with bilateral salpingo-oophorectomy coupled with sigmoid resection. In premenopausal women who want to preserve fertility, an ipsilateral salpingo-oophorectomy or salpingectomy with sigmoid resection is the treatment of choice [21]. Laparoscopy has the advantage of shorter hospitalization, compared with the classical approach, has fewer complications, but requires a longer operating time [22]. This surgical option seems to be safe, but there is lack of randomised clinical trials in the literature [24].

The conservative, nonsurgical treatment (antibiotics, standard intravenous fluids, and artificial nutrition) is an ineffective therapy and it is associated with a higher recurrence risk [25]. This management option is only reserved for the patients with comorbidities and high anesthesia risk or those who decline surgical treatment.

The novelty of this case report is that, unlike the literature reports, our patient presented with gynecological problems and her only symptom was the vaginal discharge, managed in our gynecology department.

## 4. Conclusions

Colosalpingeal fistula is a rare complication of intestinal diverticulosis, and the diagnosis should be suspected by the gynecologist in a patient with abdominal pain and known diverticular disease, especially in the presence of persistent or recurrent vaginal discharge. The purpose of our paper is to bring attention to the recurrent vaginal discharge and to make the differential diagnosis with the colotubal fistula, especially in patients with diverticular or inflammatory disease.

## Figures and Tables

**Figure 1 medicina-56-00477-f001:**
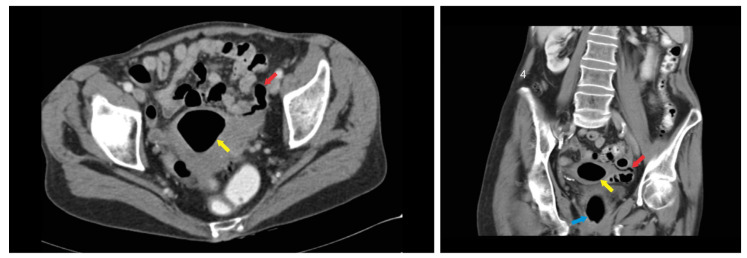
Computed tomography (CT) scan: **Left side**—axial contrast—enhanced CT scan of pelvis shows the presence of air in the uterine cavity (yellow arrow) and enlargement of left adnexa which contains gas (red arrow); **right side**—coronal contrast—enhanced CT pelvis image with rectal lumen (blue arrow), the air presence in the uterine cavity (yellow arrow), and the air in the tubal lumen (red arrow).

**Figure 2 medicina-56-00477-f002:**
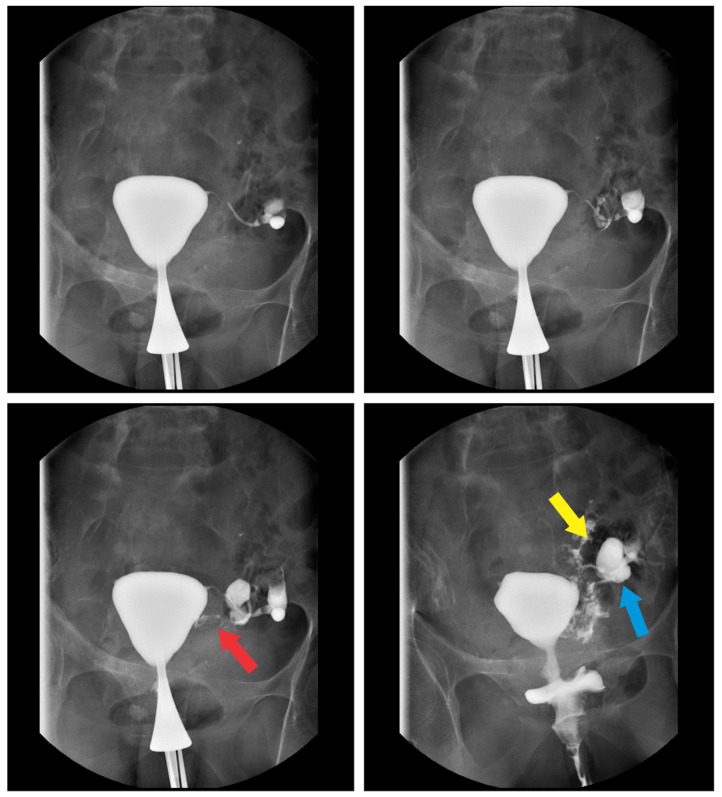
Hysterosalpingography: (red arrow)—tubal lumen; (blue arrow)—intestinal lumen; (yellow arrow)—tubal fistula.

**Figure 3 medicina-56-00477-f003:**
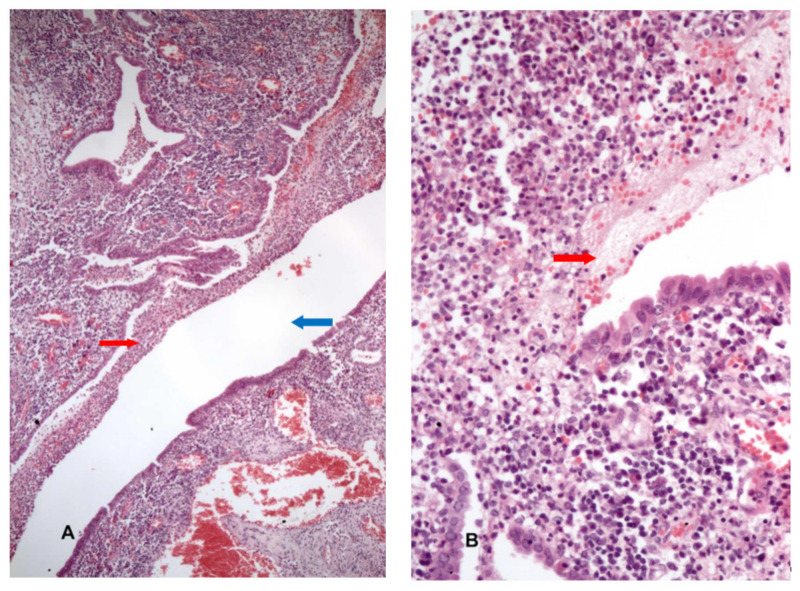
Microscopic examination: (**A**) H&E slide ×5 magnification. Chronic and subacute salpingitis (red arrow; tubal lumen—blue arrow). (**B**) H&E slide ×10 magnification. Stercoral debris mixed with extensive abscessation beaches (red arrow).

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
