# Peer review of "A Very Rare Case of Colosalpingeal Fistula Secondary to Diverticulitis: An Overview of Development, Clinical Features and Management"

_medicina, 2020, doi:10.3390/medicina56090477_

Round 1

Reviewer 1 Report

Authors:

The purpose of our paper is to bring attention in the recurrent vaginal discharge and to make the differential diagnosis with the colo-tubal fistula especially in patients with diverticular or inflammatory disease.

Comment 1:

Then, this message should be included in the conclusion section. Authors claimed that “Early treatment is essential, especially in patients who wish to preserve fertility.” as the last comment (line 158). However, the present case is 69 years old, and this report does not show the treatment strategy to preserve the fertility.

Authors:

The novelty of this case report is that, unlike the literature reports, our patient presented with gynecological problems and her only symptom was the vaginal discharge, managed in our gynecology department.

Comment 2:

Authors should show the novelty in the discussion section because that take-home message will help readers.

Author Response

 Thank you for your kind comments regarding our paper.

 The purpose of our paper is to bring attention in the recurrent vaginal discharge and to make the differential diagnosis with the colo-tubal fistula especially in patients with diverticular or inflammatory disease.

Comment 1:

Then, this message should be included in the conclusion section. Authors claimed that “Early treatment is essential, especially in patients who wish to preserve fertility.” as the last comment (line 158). However, the present case is 69 years old, and this report does not show the treatment strategy to preserve the fertility.

Thank you for your suggestion. We have added the paragraph in the recommended section and the and I deleted the comment about fertility preservation treatment.

The novelty of this case report is that, unlike the literature reports, our patient presented with gynecological problems and her only symptom was the vaginal discharge, managed in our gynecology department.

Comment 2:

Authors should show the novelty in the discussion section because that take-home message will help readers.

Thank you for your suggestion.The explanations with the novelty of our case were added in the text.

Nicolae Gica

Filantropia Hospital 11 -13, Blvd. Ion Mihalache Bucharest, Romania

E-mail:[email protected]

Tel/Fax: +40727827815

Reviewer 2 Report

Responses to editor and Reviewers' comments are clear and exhaustive.

Requested changes have been done and now the manuscript is suitable.

Author Response

Open Review

 Thank you for your comments.

Reviewer 3 Report

The authors successfully replied to the questions raised,  the manuscript can pending minor change: references have to be reported according to Journal style.

Author Response

 Open Review

The authors successfully replied to the questions raised, the manuscript can pending minor change: references have to be reported according to Journal style.

 Thank you for your comment - the references were modified according to the journal guidelines.

Round 2

Reviewer 1 Report

Thank you for giving me such an opportunity.
I agree editor and other reviewers' decision (accept).
Sincerely,

This manuscript is a resubmission of an earlier submission. The following is a list of the peer review reports and author responses from that submission.

Round 1

Reviewer 1 Report

Dr. Plopa et al. reported a case of colosalpingeal fistula caused by diverticulitis, successfully treated with one-stage surgery. I congratulate them on their success.

Major comments:

  1. What is the purpose of this case report? It seems that introduction does not include the things to be clarified by their experience.

  1. What is the novelty of this case report? Discussion section consists only of literature review. As authors mentioned in line 86, there are at least 10 case reports of colosalpingeal fistula. Furthermore, authors introduce the original article of this disease (reference No.15). According to that original article, colosalpingeal fistula affects the left tube in 93% like presented case (line 102). I think that authors shoud clarify the novelty of this case report.

Minor comments:

  1. The name of bacteria such as Escherichia Coli should be written in Italic type (line 46).

  1. How to write references does not obey journal guideline.

  1. Figure 3 is difficult to understand the details of the specimen. It will help readers to understand the figure if there is more detailed explanation with the marks such as arrows and/or arrowheads.

Reviewer 2 Report

The Authors report a rare case of colosalpingeal fistula secondary to diverticuitis. The case was well studied and it is well reported with clear imaging: The discussion is exhaustive and perhaps too long.  

Reviewer 3 Report

Although acute diverticulitis is generally uncomplicated, about 20% of those patients develop a complicated disease, and about 4% of them may develop a fistula. This interesting case report described a rare complication of acute diverticulitis, namely the occurrence of a colosalpingeal fistula. At the beginning, the management of the patient was in gynecological setting, and following diagnosis of colosalpingeal fistulas was performed after abdominal CT scan and hysterosalpingography. Finally, surgical management was made with resolution of the symptoms. 

Since colosalpingeal fistula is a rare complication of acute diverticulitis, this interesting case is worthy of publication. However, it needs  some improvement before acceptance.

  1. Paragraph 2. Please explain what is the mean of the definition “G2P2”.
  2. The recent review by Tursi et al (Nat Rev Dis Primers 2020;26:20) is a comprehensive review about the epidemiology, diagnosis and treatment of this disease. This review has to be cited;
  3. Why the patient did undergo hysterosalpingography following abdominal CT? CT is considered the gold standard to pose the diagnosis of acute diverticulitis and its complications (see again the review by Tursi et al.), and authors claimed this correctly in Discussion: please explain.

4. The discussion is too long, it must to be reduced by a third. In example, authors spent 7 rows to discuss about the role of barium enema in diagnosis colosalpingeal fistulas. However, this radiology technique is probably obsolete due to high performance of CT scan (see again the review by Tursi et al.). The Discussion should follow the following topics: the prevalence of fistulas in diverticular disease; how to pose the right diagnosis; how to perform the correct differential diagnosis; which is the best treatment and the its final outcome. This paragraph should take no more than 7-800 words
